# Postprandial Metabolic Effects of Fiber Mixes Revealed by in vivo Stable Isotope Labeling in Humans

**DOI:** 10.3390/metabo9050091

**Published:** 2019-05-07

**Authors:** Lisa Schlicker, Hanny M. Boers, Christian-Alexander Dudek, Gang Zhao, Arnab Barua, Jean-Pierre Trezzi, Michael Meyer-Hermann, Doris M. Jacobs, Karsten Hiller

**Affiliations:** 1Department for Bioinformatics and Biochemistry, BRICS, Technische Universität Braunschweig, Rebenring 56, 38106 Braunschweig, Germany; l.schlicker@tu-bs.de (L.S.); c.dudek@tu-bs.de (C.-A.D.); 2Unilever R&D Vlaardingen, Olivier van Noortlaan 120, 3133 AT Vlaardingen, The Netherlands; hanny.boers@unilever.com (H.M.B.); doris.jacobs@unilever.com (D.M.J.); 3Department of Systems Immunology and Braunschweig Integrated Centre of Systems Biology, Helmholtz Centre for Infection Research, Rebenring 56, 38106 Braunschweig, Germany; gzh11@helmholtz-hzi.de (G.Z.); arnab.barua@theoretical-biology.de (A.B.); mmh@theoretical-biology.de (M.M.-H.); 4Centre for Individualised Infection Medicine (CIIM), Feodor-Lynen-Straße 15, 30625 Hannover, Germany; 5Integrated Biobank of Luxembourg, Luxembourg Institute of Health, 1 rue Louis Rech, 3555 Dudelange, Luxembourg; jean-pierre.trezzi@ext.uni.lu; 6Luxembourg Centre for Systems Biomedicine, Université du Luxembourg, 7 avenue des Hauts-Fourneaux, 4362 Esch-sur-Alzette, Luxembourg; 7Institute of Biochemistry, Biotechnology and Bioinformatics, Technische Universität Braunschweig, Spielmannstraße 7, 38106 Braunschweig, Germany; 8Helmholtz Centre for Infection Research, Inhoffenstraße 7, 38124 Braunschweig, Germany

**Keywords:** ^13^C-enrichment, GC-MS, wheat flour, guar gum, chickpea flour, metabolism, starch

## Abstract

Food supplementation with a fiber mix of guar gum and chickpea flour represents a promising approach to reduce the risk of type 2 diabetes mellitus (T2DM) by attenuating postprandial glycemia. To investigate the effects on postprandial metabolic fluxes of glucose-derived metabolites in response to this fiber mix, a randomized, cross-over study was designed. Twelve healthy, male subjects consumed three different flatbreads either supplemented with 2% guar gum or 4% guar gum and 15% chickpea flour or without supplementation (control). The flatbreads were enriched with ~2% of ^13^C-labeled wheat flour. Blood was collected at 16 intervals over a period of 360 min after bread intake and plasma samples were analyzed by GC-MS based metabolite profiling combined with stable isotope-assisted metabolomics. Although metabolite levels of the downstream metabolites of glucose, specifically lactate and alanine, were not altered in response to the fiber mix, supplementation of 4% guar gum was shown to significantly delay and reduce the exogenous formation of these metabolites. Metabolic modeling and computation of appearance rates revealed that the effects induced by the fiber mix were strongest for glucose and attenuated downstream of glucose. Further investigations to explore the potential of fiber mix supplementation to counteract the development of metabolic diseases are warranted.

## 1. Introduction

There is an increasing worldwide prevalence of type 2 diabetes mellitus (T2DM), especially in the Asian population [1]. It is known that moderate lifestyle interventions such as a prudent diet and exercise can help to prevent T2DM [2]. In addition, lowering of the postprandial glucose response may reduce progression from pre-diabetes to T2DM [3,4,5,6]. Lowering the postprandial glucose (PPG) response of staple foods is particularly worthwhile, because of their widespread and frequent consumption [7]. The two most common carbohydrate-rich staple foods in South Asia are rice and wheat-based flatbreads [8]. In a previous study, we found that supplementation of guar gum and chickpea flour to wheat-based flatbread lowered dose-dependently the postprandial glucose and insulin responses in Indian and Caucasian subjects [9,10]. Guar gum is a water-soluble viscous fiber extracted from *Cyamopsis tetragonoloba* [11,12] and its consumption has been linked to beneficial health effects including reduced blood cholesterol, enhanced mineral absorption and stimulation of short chain fatty acid production [11,13]. It is assumed that the viscosity and gel-forming characteristics of guar gum are responsible for the reduction of the postprandial glucose response [12]. Chickpea flour is rich in high-quality protein and dietary fibers and similar to guar gum, its consumption is considered to lower postprandial glycemia and insulin [14,15]. Consequently, the ingestion of guar gum in combination with chickpea flour results in delayed gastric emptying and reduced rates of starch hydrolysis and subsequent glucose absorption [12,16,17]. But testing for this hypothesis requires information on the dynamics of glucose metabolism considering that the postprandial glucose response is the integrated result of the rate of appearance of exogenous glucose, rate of disappearance of total glucose and the endogenous glucose production in the liver [18]. Using the dual tracer approach, Boers et al. showed that guar gum supplementation reduced the rates of exogenous glucose appearance, but also had substantial effects on the rates of postprandial disposal and endogenous glucose production [19]. This suggests that the addition of the fiber mix not only affects gastric emptying and the digestion process in the intestine, as expected [16], but also impacts other metabolic rates.

Metabolomics techniques are increasingly applied in the field of nutrition to assess changes in the postprandial metabolite levels in response to a nutritional intervention [20,21]. However, changes in metabolite levels result from changing metabolic fluxes [22], therefore, knowledge of metabolic fluxes contributes to a better understanding of the mechanism behind a nutritional intervention of interest. To reveal metabolic fluxes through pathway metabolites, stable-isotope labeling can be applied. The metabolic conversion of stable-isotope labeled metabolites is almost identical to that of the respective unlabeled analogue [23], however, the molecular masses of the compound itself and of all downstream metabolites are increased and can thus be distinguished by mass spectrometry.

To date, the knowledge on postprandial metabolic fluxes is limited as nutritional metabolomics studies exclusively look at postprandial metabolite levels [24,25,26,27,28,29,30] while studies that include stable isotope labeling typically investigate the kinetic turn-over of single metabolites only, most often glucose [18,31,32,33] and highly complex dual and triple tracer approaches need to be applied to disentangle the kinetic rates that contribute to the plasma concentration of the respective metabolite [34]. To provide a better understanding of the postprandial fluxes downstream of glucose, we here combined metabolite profiling and stable isotope-assisted metabolomics to analyze the plasma samples of a nutritional intervention study assessing the effects of guar gum and chickpea flour supplementation in wheat-based flatbreads. In a randomized, cross-over study, 12 healthy subjects consumed flatbreads with three different flour compositions (Control, GG2 and GG4) where the Control was composed of 100% wheat flour while GG2 and GG4 were supplemented with 15% chickpea flour and 2% or 4% guar gum, respectively. In each flatbread, 2% of the wheat flour were substituted with fully ^13^C-labeled wheat flour. Using a GC-MS based workflow specifically designed for the quantification of low 13C-enrichment [35], we followed the catabolism of ^13^C wheat flour and determined the mass isotopomer distributions (MIDs) of twelve ^13^C-labeled plasma metabolites. The appearance rates of these metabolites were determined based on a single metabolite ordinary differential equation model. Moreover, a network model was applied to answer the question whether metabolic rates of downstream glucose metabolites differed from those of glucose. Finally, we performed non-targeted GC-MS based metabolite profiling to reveal effects on the postprandial plasma metabolome. To our knowledge, this is the first study assessing a comprehensive dataset of metabolome and metabolic flux data on the consequences of the ingestion of guar gum and chickpea flour in humans.

## 2. Results and Discussion

### 2.1. Postprandial Effect of Sharbati Flour Flatbreads on the Plasma Metabolome

To measure the postprandial metabolite profiles in plasma, we extracted plasma metabolites at 16 timepoints from volunteers who ingested Sharbati flour flatbreads and performed a non-targeted GC-MS based metabolomics analysis.

Based on the mass spectrometric measurement, we detected in total 380 metabolites in the plasma of the 12 studied persons. Using repeated measures ANOVA (rmANOVA, *p* < 0.05), we found a total of 100 metabolites showing significantly altered levels in the postprandial phase (Appendix A). Of those, we were able to identify 32 polar metabolites. Their time-resolved levels are depicted in Figure 1 and the corresponding line graphs are provided in Appendix A. As expected, the abundance of most metabolites increased within the first hour after food intake, reaching a maximum between 45 and 60 min. Among those compounds were carbohydrate-related metabolites (glucose, fructose, gluconic acid) and amino acids (such as phenylalanine, threonine, serine, tyrosine, glutamine, isoleucine, valine, glutamate, lysine, proline and glycine) resulting from the hydrolysis of starch and proteins present in the flatbread [29]. In contrast, the levels of alanine and lactate started to increase at later timepoints and reached their maximum only 75 min post ingestion. After 360 min, the levels of most metabolites decreased either back to or even below their baseline concentration. In contrast, the levels of a few metabolites including the ketone body 3-hydroxybutyrate as well as the free fatty acids palmitate, stearate and octadecenoate declined directly after flatbread intake and reached a minimum between 105 and 150 min. These metabolites are synthesized by adipose tissue and the liver under fasting conditions, however, in the postprandial state the levels of these metabolites decrease due to insulin-dependent inhibition of lipolysis [29,36]. After 360 min, the abundance of these metabolites increased to twice their baseline levels as a result of insulin depletion [29,36]. Taken together, we observed increased levels of metabolites derived from food catabolism while the amounts of endogenous energy metabolites such as ketone bodies and fatty acids were decreased directly after food intake. The observed metabolite profiles are in accordance with other studies and reflect the absorptive and the post-absorptive phase after food ingestion validating our clinical and experimental setup [27,28,29,37].

### 2.2. Incorporation of ^13^C Enrichment from 2% Labeled Wheat Flour in Central Carbon Metabolites

Next, we aimed to study to which extent plasma metabolites are derived from the ingested flatbread during digestion and catabolism. Due to the continuous turnover (appearance and disappearance) of plasma metabolites, the contribution of food derived precursors to plasma metabolite pools cannot be determined based on time-resolved concentration changes alone. To determine the fraction of the blood metabolite pool that is food-derived, a low amount (2%) of the wheat flour was substituted with fully ^13^C-labeled wheat flour during flatbread preparation. Because of the isotope incorporation, downstream metabolites can be traced and their isotopic enrichment can be quantified. Wheat flour is composed of approximately 63–72% starch and 12% protein [38], both of which are hydrolyzed by amylases and proteases in the gastrointestinal tract (GIT), resulting in the release of glucose and free amino acids, respectively, into the blood. To detect flour-derived ^13^C isotopes in these metabolites, we applied a specific GC-MS based workflow, especially developed for the sensitive detection of very low ^13^C-enrichments [35]. Since the initial enrichment in the wheat flour was only 2%, and mixing with endogenous plasma metabolites causes further dilution, we expected very low isotope enrichments in plasma metabolites.

Figure 2a depicts the atom transitions for the conversion of starch derived glucose via glycolysis and the tricarboxylic acid (TCA) cycle as well as the appearance of the free amino acids after protein hydrolysis. It is apparent that essential amino acids such as lysine and isoleucine can only be derived from protein hydrolysis whereas carbohydrates such as glucose can only be derived from starch hydrolysis. On the other hand, several intermediates of central carbon metabolism can originate from both sources. As an example, serine can be catabolically provided by protein hydrolysis or anabolically by endogenous amino acid synthesis. Both pools, anabolic and catabolic, are mixed in the blood and can, in most cases, not be distinguished. As depicted in Figure 2b, we were able to detect and quantify the ^13^C-enrichment profiles of 12 central carbon metabolites, specifically for glucose, serine, glycine, lactate, alanine, citrate, glutamate, glutamine, threonine, tyrosine, lysine and isoleucine. The stable-isotope labeling allows for pinpointing to the origin of some metabolites.

Glutamate or glutamine are either directly derived from protein hydrolysis or from starch hydrolysis and subsequent TCA metabolism of glycolytic acetyl-CoA. In the first case, all 5 carbon atoms will be ^13^C enriched and produce M5 isotopologues if labeled wheat protein is hydrolyzed. In the second case, only two carbon atoms will be replaced by stable-isotopes and produce M2 isotopologues (Figure 2a). In our dataset, glucose, lactate, citrate (M2), glutamate (M2) and glutamine (M2) originate from starch, however, for glutamine (M2), due to unknown reasons, we were unable to generate accurate MIDs (Appendix A). Glutamine (M5), glutamate (M5), threonine, tyrosine, lysine and isoleucine, on the other hand, originate from wheat protein. The amino acids serine, glycine and alanine can either originate from starch *via* glycolysis or from protein hydrolysis, both pathways will result in identical isotopologues.

After ingestion of the flatbread prepared from 2% ^13^C enriched flour, we observed a maximum of 0.98% isotopic enrichment in plasma glucose (Figure 2b, left panel). For the downstream metabolites of glucose, we found 0.74% of the lactate and 0.51% of the alanine pool to be labeled from the 2% ^13^C-enriched flour (Figure 2b, left panel), indicating that a significant fraction of glycolytic intermediates is fueled by the food product. Intriguingly, we robustly detected starch-derived enrichment even in metabolites linked to the TCA cycle such as citrate and glutamate. Due to mixing with endogenous metabolite pools, we only found 0.42% and 0.24% isotopic enrichment, respectively (Figure 2b, left panel). Assuming a linear relationship between the input of ^13^C-enrichment (2%) and the resulting enrichment, for starch-derived metabolites, we multiplied the obtained values by 50 to simulate 100% ^13^C-enriched wheat bread. At the time point of maximal enrichment, 50% of the total glucose pool, 35% of the total lactate pool and 25% of the total alanine pool was food-derived. The contributions of food-derived citrate (21%) and glutamate (12%) were further reduced, indicating that the maximal ^13^C-enrichment is decreasing as the metabolic distance to glucose increases.

Interestingly, protein hydrolysis only contributed to a quite low extent to most amino acid pools due to the lower fraction of protein (12%) in the wheat flour as compared to that of starch (77%) [38]. The isotopic enrichment was below 0.20% for the majority of them. The only exceptions were the essential amino acid isoleucine with a labeled fraction of 0.40% (20% if 100% labeling is assumed) (Figure 2b, right panel) and the glycolysis-derived amino acids alanine, serine and glycine with maximal enrichments of 0.51% (25.5% when 100% labeling is assumed), 0.14% (7% when 100% labeling is assumed) and 0.18% (9% when 100% labeling is assumed) (Figure 2b, middle panel), respectively. However, in the case of alanine, glycine and serine, labeling from starch and protein hydrolysis can mix and explain the higher enrichments in these metabolites (see Figure 2a for atom transitions).

The maximal enrichment obtained for the plasma metabolites is determined by different factors, most importantly the amount of labeling introduced into the system and the size of the non-labeled metabolite pool. The most frequent amino acid found in wheat gluten is glx, referring to the mixed fraction of glutamate and glutamine, with a relative abundance of 31 mol% (2450 µmol/g) [39]. We thus expected to observe the highest maximal enrichment of all analyzed amino acids in glutamate and/or glutamine with non-labeled plasma pool sizes of ~61 µM for glutamate and 500 to 600 µM for glutamine (Source: HMDB). Surprisingly, we only measured a very low maximal enrichment for glutamate (0.0704%) as well as for glutamine (0.073%) (Figure 2b, right panel). Instead, we observed the highest enrichment for isoleucine (0.42%), although the relative abundance of this amino acid in wheat gluten protein is only 4.1 mol% [39], whereas its plasma concentration is similar to that of glutamate (69 µM, Source: HMDB). According to Adibi and Gray, isoleucine has one of the highest intestinal absorption rates while glutamate has one of the lowest after in vivo perfusion into the jejunum of humans with equimolar amino acid mixes [40,41]. In addition, it was reported that essential amino acids are more efficiently absorbed as compared to non-essential amino acids [42]. We therefore conclude that next to the amount of introduced labeling and the non-labeled pool size, the different absorption efficiencies of dietary amino acids have a major influence on the maximum ^13^C-enrichment. From the results, we further conclude that the turnover of glutamate and glutamine is significantly higher than that of isoleucine which is in line with their different roles in metabolism. Glutamate and glutamine are important for whole body ammonium homeostasis [43,44]. Both amino acids represent cellular sinks for amino groups and are thus endogenously synthesized and secreted to the blood stream for further hepatic processing [32]. In contrast, essential branched chain amino acids are mainly prone to catabolism and feed into the TCA cycle for energy supply [45].

Taken together, we were able to record time-resolved ^13^C-enrichment profiles for 12 central carbon metabolites. We obtained higher ^13^C-enrichments in starch-derived metabolites as compared to most protein-derived amino acids representative for the composition of wheat flour. For starch-derived metabolites, the amount of enrichment decreased with increasing metabolic distance to glucose. Based on the data obtained for isoleucine, glutamine and glutamate, we concluded that the amount of incorporated ^13^C-enrichment depends on the metabolic function of the respective metabolite. The consumption of fully ^13^C-labeled wheat flour rather than a single ^13^C-labeled compound thus provides deep insights into the kinetics of the digestive processes of the major constituents of wheat flour.

### 2.3. Effects of Fiber Supplementation on Postprandial Metabolite Levels and ^13^C-Enrichment

After having determined the baseline of postprandial metabolism after chapati bread intake, we aimed to study whether guar gum and chickpea flour as food supplements have a beneficial impact on postprandial metabolism. For this purpose, 2% and 4% of guar gum (GG2, GG4) in combination with 15% chickpea flour were blended into the chapati formulation. To evaluate whether the modified formulation impacts the plasma metabolome, we started to determine the postprandial levels of plasma metabolite. Data on insulin and incretin responses were published by Boers et al. [19] and are provided in Appendix A.

In general, the impact of the fiber mix supplementation on postprandial plasma metabolite levels was only moderate when compared to control and statistical significance was not reached when using repeated measures analysis of variance (rmANOVA). However, we observed a significant reduction in the postprandial glucose level when investigating specific time points (Figure 3, upper left panel); its maximum level after 45 min was reduced by 6% in case of GG4 and by 5% in case of GG2 and the area under the curve from 0 to 120 min (AUC_0–120_) was significantly reduced for GG4 (Figure 3, upper left panel). Intriguingly, the levels of the glucose downstream metabolites lactate, alanine, citrate and glutamate were not significantly altered by the fiber addition (Figure 3).

Non-targeted GC-MS and subsequent statistical analysis of single time points further revealed a significant reduction of fructose (GG2: 13%, GG4: 6%), gluconic acid (GG2: 5%, GG4: 22%) and two unidentified metabolites (RI 2028: GG2: 6%, GG4: 12%; RI 1958: GG2: 11.6%, GG4: 11.4%) at the time point of maximal level (T_max_) (Appendix A). The free fatty acids palmitate, stearate and octadecenoate, as well as the ketone body 3-hydroxybutyrate were increased in response to the supplementation (Appendix A), most probably as a consequence of a reduced glycemic effect. The levels of 2-hydroxybutyrate, a biomarker for insulin resistance [30] and a metabolite of threonine catabolism, were increased in GG2 and GG4 (Appendix A).

While a significant impact of the fiber supplementation on the postprandial glucose response could be confirmed [19], the levels of the downstream metabolites lactate, alanine, citrate and glutamate did not significantly differ from control. We therefore expected to observe fiber mix-induced effects on the food-derived ^13^C-enrichment profile of glucose, while the profiles of downstream metabolites should be less affected.

To test for this hypothesis, we set out to analyze whether the combination of guar gum and chickpea flour resulted in different time-resolved ^13^C-enrichment profiles in the studies plasma metabolites (Figure 4). Since the isotopic enrichment originates from the ^13^C-labeled wheat flour, these graphs exclusively highlight the parts of the plasma pools that are food-derived.

We observed significant effects of GG2 and GG4 on the ^13^C-enrichment profiles of most analyzed metabolites. In line with our expectations, the response of glucose M6 isotopologues was significantly attenuated by GG2 and GG4. GG2 and GG4 delayed T_max_ by 30 and 60 min, respectively, and reduced the maximal ^13^C-enrichment by ~6% for GG2 and GG4 (*p* = 0.0015, t-dependent *p* = 0.0006) (Figure 4, upper left panel). In contrast to the metabolite levels presented in the previous section, ^13^C enrichment profiles of lactate and alanine were indeed significantly altered as a result of the fiber mix addition. In line with the dynamics of glucose M6, the formation of lactate M3 and alanine M3 was delayed by 60 min following ingestion of GG4, but not GG2. However, the maximal ^13^C enrichment of lactate M3 was reduced by 11% and 13% (*p* = 0.0194, t-dependent *p* < 0.0001) (Figure 4, upper right panel), for GG2 and GG4 respectively and for alanine M3, we observed a reduction by 13% and 17% (*p* = 0.0012, t-dependent *p* < 0.0001), respectively (Figure 4, bottom right panel). Intriguingly, these findings indicate that the reduction of the maximal enrichment caused by the interventions was stronger for lactate and alanine as compared to glucose, while the metabolite levels of lactate and alanine presented in the previous section were not affected.

The intervention-induced effects described above were less pronounced in the glucose-derived M2 isotopologues of citrate and glutamate (Figure 4, middle panel; see Figure 2 for atom transitions and isotopologue formation). Due to the increasing metabolic distance of these metabolites to the tracer, we found the maximal enrichment of these metabolites only 300 to 360 min after bread ingestion. In addition, the maximal fraction of citrate and glutamate M2 isotopologues tended to be reduced by GG2 and GG4 respectively, yet without reaching statistical significance. Similar to citrate and glutamate M2, rmANOVA did not reveal a statistically significant intervention effect on the profiles of serine M3 and glycine M2 isotopomers, although their metabolic distance to glucose is rather small. In contrast to alanine M3, which is predominantly starch-derived, protein hydrolysis is significantly contributing to serine M3 and glycine M2 ^13^C-enrichment profiles and thus represents one possible explanation for the less pronounced intervention effects observed for serine M3 and glycine M2.

Based on the obtained data, we created response curves for the protein-derived amino acids glutamate M5, glutamine M5, valine M5 and tyrosine M9 as well (Appendix A). Similar to the starch-derived metabolites, the maximal ^13^C-enrichment was reduced by GG2 and GG4, however, without a delay in T_max_. As the study was designed to investigate primarily starch hydrolysis and its downstream metabolism, the protein content and composition was not adjusted. The obtained results thus cannot answer whether the observed effect was induced by the intervention or by the different protein contents and compositions and future studies are warranted.

When comparing the metabolite levels (Figure 3) and the ^13^C-enrichment profiles (Figure 4), it must be noted that the postprandial metabolite levels return to baseline within a six hour time frame, while the labeling remains in the system for a longer period of time. The plasma level of a metabolite is the net result of exogenous production, endogenous production and consumption [18], the obtained results thus point to a fast adjustment of metabolite levels by rapid adaptation of endogenous metabolite fluxes after food intake.

### 2.4. Postprandial Metabolic Turnover of Plasma Metabolites

The integration of time-resolved metabolite levels and MIDs allows for the determination of metabolic turnover rates comprising information on postprandial kinetics and how they are affected by the interventions. We setup an ODE-based independent single metabolite model approach to compute metabolite-specific appearance rates for the different interventions Ctrl, GG2 and GG4. Due to the time-resolved blood sampling, we were able to follow the appearance of the ^13^C-enrichment in target metabolites. We assume that the metabolite is transported into the blood stream at rate k1 and disappearing from the blood stream at rate k2. We used the fraction of the respective mass isotopologues to fit the model parameters to the data and computed the respective appearance rates (Appendix A). As the different bread compositions were matched for carbohydrate but not protein content, we focused on glucose and the downstream metabolites lactate, alanine, citrate and glutamate. For the starch-derived metabolites, meaningful disappearance rates could not be computed because the profiles did not return to zero during the time range of blood collection.

The single metabolite model revealed a significant reduction in the appearance rate of glucose induced by GG4; similar trends were observed for lactate and alanine, yet without reaching statistical significance (Figure 5). While the appearance rate of glucose was reduced by 20% and 38% following GG2 and GG4 ingestion (GG2: *p* = 0.0866, GG4: *p* = 0.0173), this effect was less pronounced for lactate (GG2: ~9%, *p* = 0.0867; GG4: ~12%, *p* = 0.068) and alanine (GG2: ~5%, *p* = 0.753; GG4: ~15%, *p* = 0.084). For citrate and glutamate, the appearance rates for GG2 and GG4 did not differ from those obtained for the control. These results indicate that the effect on the overall appearance induced by the fiber addition was strongest for glucose and was attenuated downstream of glucose. We therefore conclude that a significant reduction in the ^13^C-enrichment found for lactate and alanine does not necessarily result in a reduced appearance rate. In addition, we observed a stronger effect for GG4 when compared to GG2, suggesting that the effect is dependent on the amount of ingested guar gum.

In a second modeling approach, we applied a metabolic network model to answer the question whether the effects induced by the interventions primarily follow the effect observed for glucose or if there are additional glucose-independent effects for the downstream metabolites. To this end, we designed a simplified model of central carbon metabolism to describe the generated data. The model was based on glucose, pyruvate, lactate, alanine, citrate and glutamate (Appendix A); the approach is described in detail in the Appendix A. Due to large inter-individual differences, the modeling was done per subject. Seven different models were generated. In model 1, all rates were fixed not allowing for any flexibility for intervention induced changes on any glucose-derived metabolites, and then the model parameters were fitted to the data. For the other models (2–7), a single rate per model was left flexible and the parameters were fitted again. As a quality measure, the Aikaike Information Criterion corrected for small sample sizes (AICcs) were calculated for each model and subject separately. For better comparison, the obtained AICcs were normalized to the AICc obtained for model 1 (Appendix A). Assuming that the interventions induced similar changes to downstream metabolites as to glucose, model 1 should describe the data best. In case the interventions would impact downstream rates independently, as for example the conversion from pyruvate to lactate, a model parameter will better fit with k_2f_ being kept flexible and thus resulting in lower AICcs. Lower AICcs are in this case a measure of better modeling quality. We observed inter-individual differences in the AICc profiles (Appendix A). Using this modeling approach, we found that for the majority of subjects (except subject 4 and 9), the fitting quality was strongly increased when the rates k_2b_, k_2f_ or k_3_ were left flexible (Appendix A), indicating that the intervention induced changes on the conversion rates of downstream metabolites and did not simply follow the kinetics of glucose, but were rather independent of glucose appearance. Thus, the results of both modeling approaches are in line and provided the results that the kinetics of the downstream metabolites are different from those of glucose due to the attenuation of the guar gum induced effect in the downstream metabolites of glucose.

### 2.5. Conclusions

In vivo stable isotope-assisted metabolomics in combination with metabolic profiling resulted in a comprehensive data set complementing the results of Boers et al. [19]. Boers et al. demonstrated that guar gum supplementation slightly reduced the rate of appearance of exogenous glucose, but the intervention additionally induced a reduced rate of endogenous glucose production and glucose disposal resulting in lowered glucose levels [19]. Using our GC-MS based workflow, we were able to confirm that the formation of exogenous glucose is decreased following the intervention. While the approach of Boers et al. enables to disentangle various kinetic parameters, it is limited to the analysis of one single metabolite, glucose in this case. Our approach, however, provides information on the exogenous appearance of up to 20 additional central carbon metabolites, thereby complementing the approach of Boers et al. [19].

Although we cannot directly draw conclusions on endogenous production and disposal, these processes obviously need to be adjusted in vivo in order to maintain the observed unaffected plasma levels of lactate and alanine. Since the appearance of exogenous lactate and alanine was reduced, we conclude that the intervention promotes increased endogenous production and/or reduced disposal of lactate and alanine in order to maintain constant metabolite levels. In addition and in contrast to glucose (50%), the exogenous contribution to the metabolite pools of lactate (35%) and alanine (20%) is smaller and therefore, the effect of guar gum might be counterbalanced more easily for lactate and alanine.

Based on the mechanism of action proposed for guar gum hindering the hydrolyzing enzyme α-amylase to access their substrates [12,16], we expected to see an effect on protein hydrolysis as well. Similar to enzymatic starch hydrolysis, the hydrolysis of proteins is based on proteases that need to directly interact with the protein to catalyze hydrolysis. The obtained data could not answer this question finally, thus, further investigation is required to elucidate the effects of guar gum and chickpea flour on protein hydrolysis and subsequent absorption of amino acids in humans [46,47,48].

The most important limitation of this approach is the restricted availability of fully ^13^C-labeled wheat flour due to the high production costs. In turn, the number of study subjects was limited to 12 participants and only a minor fraction of wheat flour could be substituted with its labeled counterpart. Higher fractions (>2%) would have increased the number of detectable plasma metabolite isotopologues. In this context, the enrichment in M3 glucose, a marker for gluconeogenesis [49], is of high interest, but did not exceed the limit of detection. Higher amounts of ^13^C-labeled wheat flour would, thus, extend the size of the metabolic network and metabolic fluxes could be resolved in more detail. Furthermore, we want to highlight that our approach, in contrast to the dual and triple tracer approaches, is not suitable for the determination of endogenous production and clearance rates. While the dual and triple tracer approaches provide time-point specific rates limited to one single metabolite, our approach generates data on the exogenous appearance of multiple ^13^C-enriched metabolites simultaneously. In our case, one single rate of appearance per metabolite reflects the overall exogenous appearance of ^13^C-labeling in the plasma. It is important to note that for our analysis, only the oral tracer is required, which simplifies the experimental setup. In result, the two approaches yield complementary information: the dual and triple label techniques yield very detailed information on the kinetics of one specific metabolite while this approach provides a broader overview on the exogenous appearance kinetics of multiple metabolites. The generated data demonstrate that the application of stable isotope-assisted metabolomics in combination with metabolic profiling represents a powerful tool to better understand the postprandial changes of metabolite levels and metabolic fluxes in response to a nutritional intervention.

In summary, we demonstrated that flatbread digestion and catabolism mainly fuel plasma glucose, while for its downstream metabolites lactate, alanine, citrate and glutamate, the contribution from exogenous sources is diluted as a function of their metabolic distance to glucose. This demonstrates that the major flux of flatbread nutrients towards tissues is mediated mainly *via* glucose, whereas downstream metabolite pools are only indirectly fueled by tissue sources upon glucose conversion. A guar gum intervention not only delayed glucose kinetics, but also attenuated the formation of lactate and alanine from exogenous sources. Furthermore, we demonstrated that the exogenous contribution and utilization of the investigated plasma amino acids strongly varies reflecting their different functions in human metabolism. Based on the comprehensive data set generated in the course of this study, we conclude that the main effect induced by the intervention manifests in glucose. Thus, the addition of fiber mixes to wheat-based flatbread could represent a promising strategy to lower postprandial glycemia following the consumption of wheat-based flatbreads. Further studies investigating the fiber mix addition in the context of health promotion and T2DM prevention are warranted.

## 3. Materials and Methods

### 3.1. Study Design

The complete study design including the preparation of the test products has already been described earlier [19].

#### 3.1.1. Subjects

In short, fifteen healthy men were recruited for screening from on existing database of Quality Performance Service Netherlands B.V., Groningen, The Netherlands, a clinical research organization (CRO). Twelve subjects completed the study. For inclusion/exclusion criteria and ethical aspects see Boers et al. [19]. The trial was registered at www.clinicaltrials.gov as NCT01734590.

#### 3.1.2. Experimental Design

The study used a double-blind, randomized, controlled, full cross-over (within subject) design. Subjects attended the initial screening day followed by 3 test days, at least 1 week apart. Participants were instructed to minimize changes in their habitual diet and activity during the study period and they were not allowed to consume ^13^C-enriched food (for list see reference [19]). In addition, they also got a standardized evening meal at the CRO. All participants fasted overnight (from 19.30 h until consumption of the test product) but were allowed to drink water *at libitum*. Because this was a dual labeled isotope study, next to the labeled food, a priming dose of ^2^H-labelled glucose was administered as a bolus followed by a continuous infusion of deuterated glucose for 8 h. At 2 h after the start of the infusion each morning at each test day, subjects consumed three freshly made flatbreads with 250 mL of mineral water as breakfast.

#### 3.1.3. Test Product and Preparation

The flatbread named GG2 contained 2% guar gum, 3% barley flour and 15% chickpea flour, while the so called GG4 flatbread contained 4% guar gum and 15% chickpea flour. The control flatbread was 100% based on wheat flour. Wheat seeds were intrinsically labeled by growing the wheat plants after germination in a labeling facility under an atmosphere containing ^13^CO_2_ (>97 atom% ^13^C). The test products obtained a final ^13^C-enrichment of approximately 2% atom percent excess (APE). The study was designed to investigate the metabolism of starch-derived metabolites, thus the ratio between the amount of introduced ^13^C-flour and total carbohydrate content was adjusted in the three conditions control, GG2 and GG4. The treatments were not matched for total protein and amino acid composition.

#### 3.1.4. Sample Collection

Blood samples were collected in K_2_-EDTA-tubes containing dipeptidyl peptidase-IV (BD Diagnostics) at time points −30, −5, 15, 30, 45, 60, 75, 90, 105, 120, 150, 180, 210, 240, 300 and 360 min and centrifuged at 1300 *g* for 10 min at 4 °C within 30 min after collection to limit pre-analytical variation [19]. Resulting plasma was stored at −80 °C prior extraction.

### 3.2. Analytical Techniques

#### 3.2.1. Metabolite Extraction from Plasma

Prior to metabolite extraction, plasma samples were thawed on ice for 30 min. Extraction was performed in technical triplicates. We combined 20 µL plasma with 180 µL extraction fluid composed of 4 parts of methanol and 1 part of water which contained three internal standards (Pentanedioic acid-d6 (final concentration: 20 µM), U^13^C-Ribitol (final concentration: 12.7 µM), Norleucine (final concentration: 10 µM)). The mixture was subsequently vortexed at 4 °C and 1400 rpm for 5 min and centrifuged at 4 °C and 21.000 *g* for 10 min. After centrifugation, 140 µL of the supernatant were transferred to a glass vial with insert and left-over extract was collected, mixed and used for the preparation of pool samples. All pipetting steps during metabolite extraction were performed by a pipetting robot (Freedom Evo, Tecan Life Sciences, Männedorf, Switzerland). The extracts were dried overnight at −4 °C in a refrigerated vacuum concentrator (Labconco, Kansas City, KS, USA) and stored at −80 °C prior GC-MS measurement.

#### 3.2.2. Metabolite Derivatization and GC-MS Measurement

Sample derivatization and GC-MS measurement were performed according to Krämer et al. [35]. Briefly, metabolites were derivatized with 15 µL of 2% methoxyamine hydrochloride in pyridine for 90 min and 15 µL N-methyl-N-trimethylsilyl-trifluoroacetamide (MSTFA) for additional 30 min at 40 °C under continuous shaking using a Gerstel MPS.

For GC-MS analysis, an Agilent 7890A GC with a 20 m DB-5MS capillary column (0.18 mm inner diameter, 0.18 µm film thickness) connected to an Agilent 5975C inert XL MSD was used. Details of the GC-MS measurement are provided elsewhere [35].

### 3.3. Data Analysis

Obtained SIM and SCAN GC-MS data were processed using the software package MetaboliteDetector [50] as described by Krämer et al. [35].

#### 3.3.1. MID Determination

MIDs were determined from the generated SIM data using MetaboliteDetector’s sum formular-based MID wizard [50]. For the mass isotopomers of the metabolites glycine, serine, citrate and glutamate, the non-labeled spectra based correction was applied [51].

#### 3.3.2. Batch Quantification and Normalization

We applied MetaboliteDetector’s batch quantification wizard for the relative quantification of the metabolite levels for the obtained SCAN data. Metabolite levels were subsequently normalized to pool samples as specified elsewhere [52].

The collected SIM glucose data did not allow for accurate MID determination, thus, the glucose M6 data were kindly provided by the Department of Laboratory Medicine, University Medical Center Groningen, University of Groningen, Groningen, The Netherlands. Plasma glucose levels were derived from the collected SCAN data.

#### 3.3.3. Contribution of Food-Derived Metabolites to the Total Metabolite Pool

The maximal contribution of food to the plasma metabolite pool was calculated based on the maximal ^13^C-enrichments obtained after the consumption of the control flatbread enriched by 2%. We assumed a linear relationship between the ingested and the obtained ^13^C-enrichment [53]. To determine the contribution of food to the plasma metabolite pool, the obtained enrichment was multiplied by 50 to simulate the ingestion of flatbread enriched by 100%.

#### 3.3.4. AUC Determination

AUC_0–120min_ of the metabolite levels of glucose, lactate, alanine, citrate and glutamate were calculated with R (version 3.4.4) using the package “DescTools” with the trapezoidal rule.

### 3.4. Statistical Analysis

#### 3.4.1. Outlier Analysis

Outliers were identified using principal component analysis (PCA) in SIMCA 15.0 (Umetrics, Umea, Sweden). In the untargeted dataset, 76 out of 1661 measurements were identified as outliers and thus excluded for further analysis. In the MID dataset, 679 (out of 1190 measurements × 115 MIDs = 136,850) single data points were excluded.

#### 3.4.2. Repeated Measures (rm)ANOVA

Repeated measures ANOVA (rmANOVA) was performed using R version 3.4.4. For both, the untargeted data set and the MID data set, the analysis was performed on the medians of the technical triplicates on the factors intervention and time and the combination of both. We accounted for multiple testing using the Benjamini & Hochberg *p*-value adjustment [54]. After *p*-value adjustment, a value of *p* < 0.05 was considered statistically significant.

#### 3.4.3. Statistical Analysis of Single Time Points

Analysis of variance (ANOVA) was performed using JMP Pro 14.0.0 (SAS Institute Inc. 2013, Cary, NC, USA). For the untargeted data set, mixed models were built per time point using the fixed factors treatment (Control, GG2, GG4), replicate and baseline (median of triplicate measurement at time points −30 and −5 min) and the random factor subjects. For the MID data set, the same mixed models were built, yet without the fixed factor baseline. Post-hoc multiple comparisons of the LSMeans using a Dunnett-Hsu adjustment were compared to compare the effects of GG2 and GG4 to control, respectively. A value of *p* < 0.05 was considered statistically significant.

### 3.5. Metabolic Modeling

#### Single Metabolites Model

We set up a single metabolite model to compute the appearance and the disappearance rates of the labeled metabolites separately. We, thereby, assume that the ^13^C-labeled metabolite is transported from a precursor compartment (*metabolite_13Ccomp_*) e.g. the intracellular compartment, into the blood stream (*metabolite_13Cblood_*) at rate k_1_ (appearance rate) and leaving the blood stream at rate k_2_ (disappearance rate) upon consumption. We assume the pool size of the ^13^C-labeled metabolite in the precursor compartment (*metabolite_13Ccomp_*) to decrease when it is transported into the blood stream (*metabolite_13Cblood_*). The model can be described using the following system of ordinary differential equations (ODEs) (Equation (1)).
(1)$$\begin{array}{c} \frac{dmetabolite_{13Ccomp}}{dt}=-k_{1}\cdot metabolite_{13Ccomp}\\ \frac{dmetabolite_{13Cblood}}{dt}=k_{1}\cdot metabolite_{13Ccomp}-k_{2}\cdot metabolite_{13Cblood} \end{array}$$

We used the model described above to compute the appearance rate (k_1_) and disappearance rate (k_2_) for metabolites that incorporated ^13^C labeling. For fitting, we used the relative metabolite levels multiplied with the mass isotopomer (MI) abundance specifically enriched from starch or protein hydrolysis in percent resulting in relative amounts of labeling. The parameters of the model are estimated using the Levenberg-Marquardt [55,56] algorithm for least squares curve fitting. To overcome the initial value problem of the method we used a genetic algorithm to solve the system with different, randomly generated initial values. As a result, we received a set of optimized parameters with the corresponding root mean square deviation (RMSD) for every run of the genetic algorithm (Equation (2)).
(2)$$RMSD=\sqrt{\frac{\sum_{t=1}^{N}{\left(\hat{y_{i}}-y_{i}\right)}^{2}}{N}}$$

The final parameters are the means of the best fitting parameter sets. We used the parameters of the control to compare with the different conditions using the two-sided paired t-test. A *p*-value < 0.05 was considered significantly different. To compare the effect size between metabolites, the computed rates for GG2 and GG4 were normalized to the respective Ctrl rate and multiplied by 100 to obtain a percentage.

## Figures and Tables

**Figure 1 metabolites-09-00091-f001:**
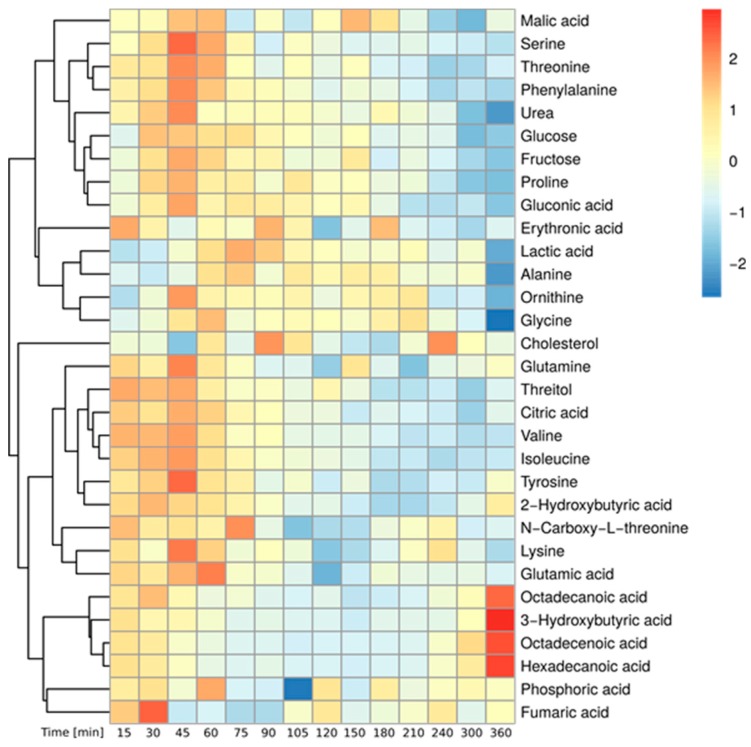
Postprandial metabolite levels after wheat bread intake—Heatmap of identified, significantly changed metabolites over time (Median of all subjects, Baseline correction, zScore Normalization, rmANOVA: *p* < 0.05, adjusted for multiple comparison by Benjamini & Hochberg procedure).

**Figure 2 metabolites-09-00091-f002:**
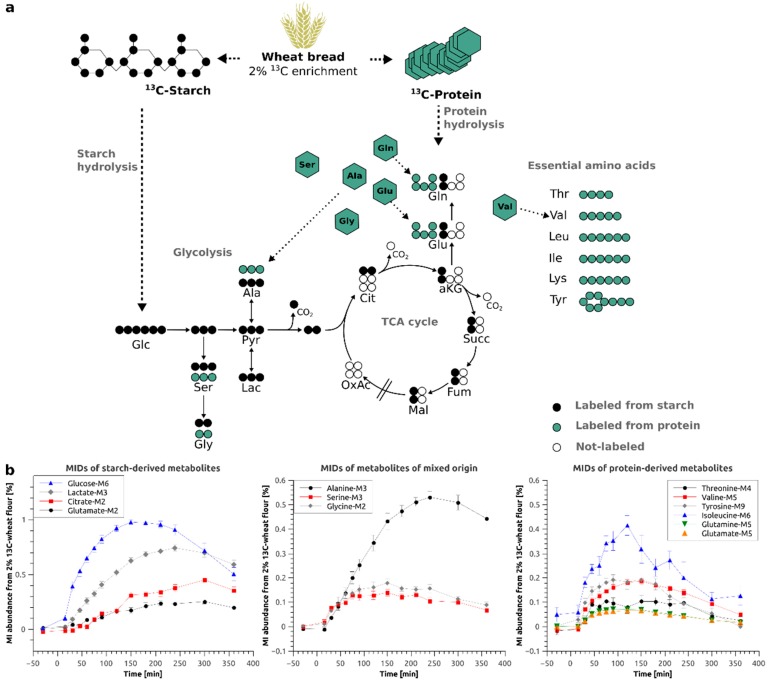
(**a**) Schema of wheat bread digestion (starch and protein hydrolysis) and atom transitions for metabolic conversions of glucose *via* glycolysis and TCA cycle; (**b**) Time-resolved mass isotopomer (MI) abundances of starch-derived (left) and protein-derived (right) metabolites and metabolites of mixed origin (middle) in %–Average of all subjects ± Standard Error; Glc–glucose, Ser–serine, Gly–glycine, Pyr–Pyruvate, Ala–alanine, Lac–lactate, Cit–citrate, aKG–α-ketoglutarate, Succ–succinate, Fum–fumarate, Mal–malate, OxAc–oxaloacetate, Glu–glutamate, Gln–Glutamine, Thr–Threonine, Val–Valine, Leu–Leucine, Ile–Isoleucine, Lys–Lysine, Tyr–Tyrosine.

**Figure 3 metabolites-09-00091-f003:**
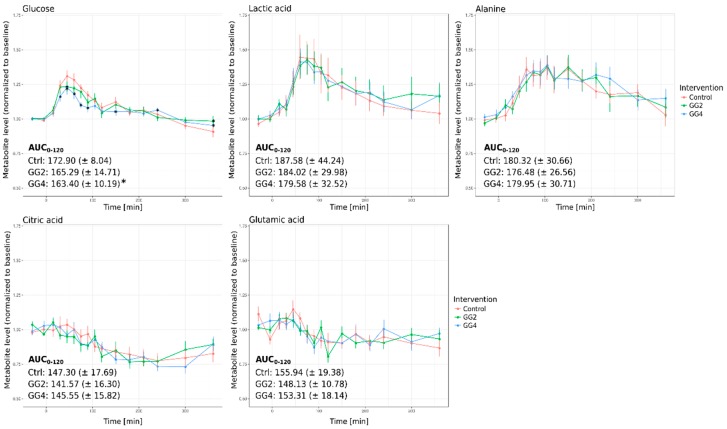
Relative metabolite levels over time and AUC_0–120_ for glucose, lactic acid, alanine, citric acid and glutamic acid (Red–Control, Green–GG2, Blue–GG4; average metabolite levels normalized to baseline ± standard error, * indicates significant differences at single time points in relation to Ctrl, *p* < 0.05).

**Figure 4 metabolites-09-00091-f004:**
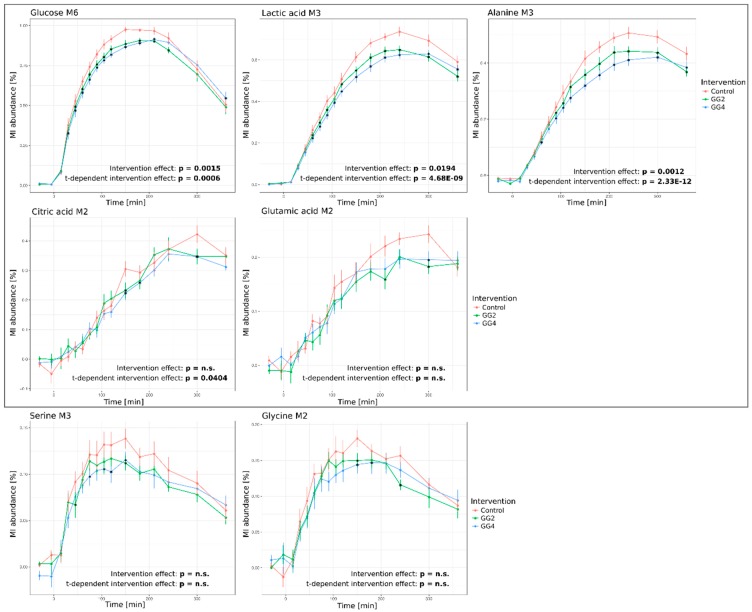
Response curves of ^13^C-enrichment profiles over time upon intervention for glucose M6, lactate M3, alanine M3, citrate M2, glutamic acid M2, serine M3 and glycine M2 (Red–Control, Green–GG2, Blue–GG4; Average of MI abundance in % ± standard error of 12 subjects; grey box frames metabolites included for modelling; combined statistical analysis using rmANOVA (stated *p*-values) and analysis of single time points (indicated by *), *p* < 0.05, adjusted for multiple comparison by Benjamini & Hochberg procedure).

**Figure 5 metabolites-09-00091-f005:**
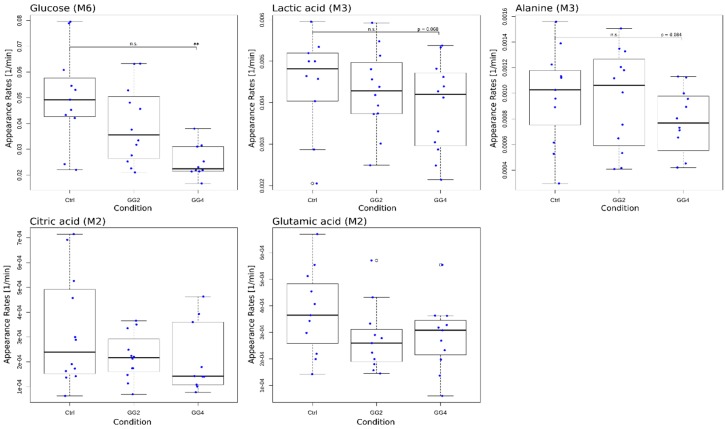
Boxplots of modeled appearance rates for glucose M6, lactate M3, alanine M3, citrate M2 and glutamate M2 with single data points in blue; * indicates significant differences in relation to Ctrl, *p* < 0.05.

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
