# Peer review of "Postprandial Metabolic Effects of Fiber Mixes Revealed by in vivo Stable Isotope Labeling in Humans"

_metabolites, 2019, doi:10.3390/metabo9050091_

Round 1

Reviewer 1 Report

- At the beginning of the 3rd paragraph in the introduction, it is stated that stable isotope tracers can be used to study metabolite kinetics, with glucose being one of the most studied metabolites. I feel the authors could make mention of the complexity required to study glucose kinetics in the postprandial state by providing some detail about the dual and triple tracer approaches. Robert Rizza has a couple of recent reviews/perspectives on this which have been published in Diabetes. Also in this regard, a limitation of this approach is the fact that kinetic parameters of all aspects of glucose turnover were not able to be determined, as can be with the triple tracer method. This should be discussed in the conclusions. 

- Consideration of the accessibility of this method to those wanting to establish this or similar analytic workflows should be discussed. For example, what is the cost on a per person basis for the 13C-labeled wheat flour, what the technical demands for detecting low enrichment, etc? ie. is this a feasible approach for the average lab to establish?      

- Presenting Figure 1 as a heat map of metabolite levels is not readily accessible by the reader and is uninformative. Consider using line graphs as this may convey this data in a more meaningful way.

- Data on the actual plasma glucose levels, insulin, C-peptide and total free fatty acids have not been presented. It would be beneficial to include this data.

- in the methods (section 3.1.2) it is stated that 2H-labeled glucose was infused throughout the study. Yet no tracer data is presented. Why? 

Author Response

Reviewer 1

We thank Reviewer 1 for taking the time to carefully review our manuscript. We feel that the comments are very helpful in order to improve the quality and readability of the manuscript. We provide the point-by-point responses next to the comments and all modifications in the manuscript are highlighted in blue. We hope that Reviewer 1 satisfied with the answers provided below.  

At the beginning of the 3rd paragraph in the introduction, it is stated that stable isotope tracers can be used to study metabolite kinetics, with glucose being one of the most studied metabolites. I feel the authors could make mention of the complexity required to study glucose kinetics in the postprandial state by providing some detail about the dual and triple tracer approaches. Robert Rizza has a couple of recent reviews/perspectives on this which have been published in Diabetes. Also in this regard, a limitation of this approach is the fact that kinetic parameters of all aspects of glucose turnover were not able to be determined, as can be with the triple tracer method. This should be discussed in the conclusions.”

Answer: We agree that we missed out to add an explanation of the dual and triple tracer approaches in comparison to the approach applied in this manuscript. We changed the introduction according to the comment of reviewer 1 (p. 2) and added a direct comparison between the dual/triple label approaches as compared to our appraoch in the conclusion section (p. 12-13).

Consideration of the accessibility of this method to those wanting to establish this or similar analytic workflows should be discussed. For example, what is the cost on a per person basis for the 13C-labeled wheat flour, what the technical demands for detecting low enrichment, etc? ie. is this a feasible approach for the average lab to establish?”

Answer: We agree that the accessibility of the applied method is important to discuss in the manuscript and modified our conclusion section accordingly (p. 12, bottom). Reviewer 1 is correct that the high costs for the production of the fully 13C-labeled wheat flour highly limits accessibility. Importantly, the detection of enrichment as low as 0.01% can be done with a standard single quadrupole GC-MS instrument that is accessible to many laboratories[1]⁠.

Presenting Figure 1 as a heat map of metabolite levels is not readily accessible by the reader and is uninformative. Consider using line graphs as this may convey this data in a more meaningful way.”

Answer: We thank Reviewer 1 for this comment. We also discussed a lot about the best way of presenting these data. We decided for the heatmap as one can directly see what metabolites behave similarly over time and thus cluster together; this important information would be lost when providing only line graphs. However, we see his point and now additionally provide line graphs for all metabolites presented in the heatmap in the supplementary figure S2.  

Data on the actual plasma glucose levels, insulin, C-peptide and total free fatty acids have not been presented. It would be beneficial to include this data.”

Answer: We thank Reviewer 1 for pointing this out and we agree on the importance of these parameters. Our study follows up on Boers et al.[2]⁠, who applied the dual label approach to compute the different glucose kinetics and also recorded most of the parameters mentioned above (glucose levels, insulin, GLP-1 and GIP responses). We provide these data now in Suppl. Table S1. Unfortunately, data on C-peptide were not recorded. Instead of providing the total free fatty acids, we report the data of single free fatty acids like octadecanoic acid, octadecenoic acid and hexadecanoic acid in the heatmap (Fig. 1) and the line graphs (Suppl. Fig. S2).

in the methods (section 3.1.2) it is stated that 2H-labeled glucose was infused throughout the study. Yet no tracer data is presented. Why?”

Answer: We agree with Reviewer 1 that this is confusing to the readers of the manuscript. The 2H-labeled glucose was required for the dual label approach published on this nutritional intervention study by Boers et al[2]⁠. As we investigated the same plasma samples, we need to mention in the material section that 2H-labeled glucose was infused. However, as we were only interested in the13C-enrichment coming from the labeled wheat flour, we did not report on the 2H-labeling. We now clarify in the conclusion section, that, for our approach, only the oral tracer is required (p. 13, line 2).

References

[1]     Krämer, L.; Jäger, C.; Trezzi, J.-P.; Jacobs, D. M.; Hiller, K. Quantification of Stable Isotope Traces Close to Natural Enrichment in Human Plasma Metabolites Using Gas Chromatography-Mass Spectrometry. Metabolites 2018. https://doi.org/10.3390/metabo8010015.

[2]     Boers, H. M.; Van Dijk, T. H.; Hiemstra, H.; Hoogenraad, A. R.; Mela, D. J.; Peters,             H. P. F.; Vonk, R. J.; Priebe, M. G. Effect of Fibre Additions to Flatbread Flour             Mixes on Glucose Kinetics: A Randomised Controlled. Br. J. Nutr. 2017, 118                     (10), 777–787. https://doi.org/10.1017/S0007114517002781.

Reviewer 2 Report

1.      The author claims “the metabolic conversion of stable-isotope labeled metabolites is almost identical to that of the respective unlabeled analogue”. Is there any difference? Can you find a reference about this?

2.      Paragraph 3.1.1 Subjects. Why only men were recruited? What are the baseline characteristics of the subjects? Why was the study limited to twelve subjects only?

3.      Paragraph 3.1.4 Sample collection. Plasma was stored at -20oC prior extraction. Usually plasma is stored at -80oC. How long was the time between sample collection and metabolite extraction?

4.      Paragraph 3.2.1 Metabolite extraction from plasma. There is no mention of a protein precipitation step. Was it done?

5.      Note: Reference 2, author name is Jaakko Tuomilehto, not Aakko Uomilehto.

Author Response

Reviewer 2

We highly appreciate that Reviewer 2 took the time to carefully review our manuscript and we feel that the comments really help to further improve the quality and readability of the manuscript. We provide point-by-point responses to the comments below and highlight the changes made in the manuscript in blue. We hope that Reviewer 2 issatisfied with the answers provided below.  

1. The author claims “the metabolic conversion of stable-isotope labeled metabolites is almost identical to that of the respective unlabeled analogue”. Is there any difference? Can you find a reference about this?”

We thank Reviewer 2 for pointing out that a reference was missing at this point. We included a reference (23) by Wasylenko and Stephanopoulus that focusses on the kinetic isotope effect and its influence on intracellular 13C labelling pattern [1]⁠. The kinetic isotope effect is based on the fact that the energy required to break a 13C-13C bond is higher as compared to a 12C-12C bond. As a result, the kinetic isotope effect is more pronounced for enzymes that catalyze carbon rearrangements. In this case, however, this weak effect can be neglected, as we compare between different conditions and the kinetic isotope effect is similar in the different conditions.

2. Paragraph 3.1.1 Subjects. Why only men were recruited? What are the baseline characteristics of the subjects? Why was the study limited to twelve subjects only?”

We agree with Reviewer 2 that it would have been very interesting to study a higher number of participants and also female subjects. However, the costs for the production of the 13C-labeled wheat flour are very high and therefore strongly limit the number of participants. These low number of subjects can be found in all studies that include stable isotope labeled food products:

Eelderinket al.(2011), Journal of Nutrition:10 subjects [2]

Boset al.(2003), Human Nutrition and Metabolism: 8 subjects per group [3]

Nuttallet al.(2016), Nutrition and Metabolism: 7 subjects [4]

Moreover, many studies includeonly male subjects as they show a more stable metabolite profile as compared towomen due to hormonal changes.As this study is following up on Boers et al, using plasma samples of the identical study, more detailed information on the inclusion citeria of the study can also be found in Boers et al[5].

3. Paragraph 3.1.4 Sample collection. Plasma was stored at -20oC prior extraction. Usually plasma is stored at -80oC. How long was the time between sample collection and metabolite extraction?”

We thank Reviewer 2 for pointing out this mistake, plasma was stored at -80°C prior metabolite extraction. We changed the method section accordingly (page 14, section 3.1.4).

4. Paragraph 3.2.1 Metabolite extraction from plasma. There is no mention of a protein precipitation step. Was it done?”

We thank Reviewer 2 for this question. The plasma was extracted using a mixture of methanol (4 parts) and water (1 part). The methanol enables the precipitation of the plasma proteins, which are subsequently pelleted by centrifugation. We tested various extraction methods beforehand and the one applied here was suited best for the extraction of polar metabolites while still precipitating the proteins[6]⁠.

5. Note: Reference 2, author name is Jaakko Tuomilehto, not Aakko Uomilehto.”

We thank Reviewer 2 for pointing out this mistake. We correctedthe reference now.

References

[1] Wasylenko, T. M.; Stephanopoulos, G. Kinetic Isotope Effects Significantly Influence Intracellular Metabolite 13C Labeling Patterns and Flux Determination. Biotechnol. J. 2013, 8 (9), 1080–1089. https://doi.org/10.1002/biot.201200276.

[2] Eelderink, C.; Moerdijk-Poortvliet, T. C. W.; Wang, H.; Schepers, M.; Preston, T.; Boer, T.; Vonk, R. J.; Schierbeek, H.; Priebe, M. G. The Glycemic Response Does Not Reflect the In Vivo Starch Digestibility of Fiber-Rich Wheat Products in Healthy Men. J. Nutr 2012,142, 258–263. https://doi.org/10.3945/jn.111.147884.

[3] Bos, C.; Metges, C. C.; Gaudichon, C.; Petzke, K. J.; Pueyo, M. E.; Morens, C.; Everwand, J.; Benamouzig, R.; Tomé, D. Postprandial Kinetics of Dietary Amino Acids Are the Main Determinant of Their Metabolism after Soy or Milk Protein Ingestion in Humans. J. Nutr.2003, 133 (5), 1308–1315. https://doi.org/10.1093/jn/133.5.1308.

[4] Nuttall, F. Q.; Almokayyad, R. M.; Gannon, M. C. The Ghrelin and Leptin Responses to Short-Term Starvation vs a Carbohydrate-Free Diet in Men with Type 2 Diabetes; a Controlled, Cross-over Design Study. Nutr. Metab. (Lond). 2016, 13 (1), 47. https://doi.org/10.1186/s12986-016-0106-x.

[5] Boers, H. M.; Van Dijk, T. H.; Hiemstra, H.; Hoogenraad, A. R.; Mela, D. J.; Peters, H. P. F.; Vonk, R. J.; Priebe, M. G. Effect of Fibre Additions to Flatbread Flour Mixes on Glucose Kinetics: A Randomised Controlled. Br. J. Nutr. 2017, 118 (10), 777–787. https://doi.org/10.1017/S0007114517002781.

[6] Krämer, L.; Jäger, C.; Trezzi, J.-P.; Jacobs, D. M.; Hiller, K. Quantification of Stable Isotope Traces Close to Natural Enrichment in Human Plasma Metabolites Using Gas Chromatography- Mass Spectrometry. Metabolites 2018. https://doi.org/10.3390/metabo8010015.

p { margin-bottom: 0.25cm; line-height: 115%; }
